# Comparison of Interactive Teaching in Online and Offline Platforms among Dental Undergraduates

**DOI:** 10.3390/ijerph19063170

**Published:** 2022-03-08

**Authors:** Deepak Nallaswamy Veeraiyan, Sheeja S. Varghese, Arvina Rajasekar, Mohmed Isaqali Karobari, Lakshmi Thangavelu, Anand Marya, Pietro Messina, Giuseppe Alessandro Scardina

**Affiliations:** 1Department of Prosthodontics, Saveetha Dental College and Hospitals, Saveetha Institute of Medical and Technical Sciences, Chennai 600077, India; drdeepaknallu@gmail.com; 2Department of Periodontology, Saveetha Dental College and Hospitals, Saveetha Institute of Medical and Technical Sciences, Chennai 600077, India; arvinar.sdc@saveetha.com; 3Department of Conservative Dentistry & Endodontics, Saveetha Dental College and Hospitals, Saveetha Institute of Medical and Technical Sciences, Chennai 600077, India; 4Conservative Dentistry Unit, School of Dental Sciences, Universiti Sains Malaysia, Kota Bharu 16150, Malaysia; 5Department of Pharmacology, Saveetha Dental College and Hospital, Saveetha Institute of Medical and Technical Sciences, Saveetha University, Chennai 600077, India; Lakshmi@saveetha.com; 6Department of Orthodontics, Faculty of Dentistry, University of Puthisastra, Phnom Penh 12211, Cambodia; amarya@puthisastra.edu.kh; 7Department of Surgical, Oncological and Stomatological Disciplines, University of Palermo, 90133 Palermo, Italy; pietro.messina01@unipa.it

**Keywords:** dentistry, digital education, education, e-learning, interactive teaching, online teaching

## Abstract

In recent years, the educational system has focused more on the holistic development of an individual. Modern technology has changed the educational environment to provide students with better academic opportunities. Along with the education system, teaching techniques and learning tools have also changed with digital evolution. This research was undertaken to assess the academic performance of interactive teaching methods in offline and online platforms in Periodontics among BDS undergraduates at a dental college in India. This prospective study was conducted among 49 students: Group I (n = 24, online class through Zoom) and Group II (n = 25, offline classes). The subject was divided into three modules and was covered in one week. The topics covered, teaching methods, lectures, and activities were similar for both groups. A formative assessment mark was obtained from written tests during the module, whereas the summative assessment mark was recorded from exams conducted towards the end of the module. In the results, a statistically significant difference was not observed in terms of formative assessment between Group I (77.88 ± 12.89) and Group II (77.80 ± 16.09) (*p* = 0.98). In addition, a statistically significant difference was not observed in terms of summative assessment between Group I (80.54 ± 8.39) and Group II (80.28 ± 11.57) (*p* = 0.93). Overall, this study suggests that interactive teaching methods in both offline and online platforms in Periodontics showed equivalent performance by the undergraduate dental students.

## 1. Introduction

The education system has evolved dramatically over the years. Classroom-based education systems existed for many years until the modern education system came into the picture in the nineteenth century. The fundamental strength of this modern education system is a well-defined and structured curriculum that gives importance to all the subjects. However, it does not focus on the holistic development of an individual. In addition, it is not easy to create a customized study plan to meet the needs of each individual [1].

Along with the education system, teaching techniques and learning tools have also changed with digital evolution. Small group learning, rather than traditional lecture-based learning, is incorporated into the current educational system, which is student-centered. This learning method helps the students express and communicate their thoughts with their peers [2]. In addition, to gain the students’ attention, topics are broken down into microlectures and are coupled with activities such as mind mapping, critical pedagogy, and role play. This keeps students engaged and active throughout the session. This didactic and non-didactic method promotes learning and the application of concepts, as well. The primary tactic of this interactive model of learning method should be to exemplify the performance of every student and inculcate exploratory and innovative thinking with flexible training programs in which students can learn at their own pace [3].

An interactive teaching style is a learning activity in which students participate in the process of learning and reflect on what they know, think, and believe. Against conventional methods of teaching, which place a premium on the instructors’ prominent role in assisting and facilitating students, the interactive mode of teaching places a premium on the students’ abilities and interests [4]. In the conventional classroom method, the instructor is the focus in the learning process, and students are just recipients, but in a student-centered system, the instructor and the student exchange roles, allowing the student to participate diligently in the process of learning, and they become the main focus. The main aim of the instructor or tutor in an interactive mode of learning is to help students achieve their goals. Here, the instructor creates a plan that includes activities, discussions, and problem-solving tasks that allow students to acquire new ideas and change an individual into a group task. Each person in the group contributes to the overall success of the group. The essential components of interactive lessons are interactive activities and tasks that students achieve. Therefore, this method ensures the complete involvement of students throughout learning [5]. The assignments provided in the interactive sessions also help the participants gain knowledge and make them competent enough to complete it with innovative ideas. In addition, the interactive mode of teaching ensures that every participant is actively engaged in intellectual development and everyone offers and shares their opinions, ideas, and information [6].

Furthermore, this allows students to develop multiple skills, such as listening to others, teamwork, analyzing diverse points of view, discussing, and decision making. According to the literature, interactive learning aids the student in acquiring information and retaining it for a prolonged period. It also activates students’ creative thinking and analytic and syllogistic skills, allowing them to make reasonable decisions in any situation in order to develop the most acceptable models of thinking, action, and communication [7].

E-learning is becoming more popular in many higher education institutions recently. Both learners and educators are drawn to the benefits of e-learning, which include the ability to learn anywhere, at any time, and at one’s own pace. E-learning is distributing educational information to students who are separated by a significant distance from their instructors or teachers. It makes use of the Internet, computers, networks, and multimedia technology [8]. When it comes to e-learning, it is common to use different methods. Learning through this mode can hold the students’ attention because it frequently includes interactive images, texts, audio, videos, collaborative sharing, and other features. This also permits interactive learning. Furthermore, we may access information from anywhere and anytime as long as we have a computer and an internet connection. E-learning has the potential to improve educational and training access and teaching and learning quality. It also emphasizes the importance of higher education institutions maintaining a competitive edge in this constantly shifting student market. In addition, here, technology has been fully utilized in improving the teaching and learning process while also allowing for the delivery of educational programs to a more significant number of students at a lower cost. As a result, e-learning can enhance the quality of teaching and learning [9].

The progress of society and the impact of technology are directly related to the quality of education. Different students respond to various instructional methods. Some people learn better by seeing things, while others prefer reading or listening to lectures. To combat this, teachers provide students with various possibilities and routes to comprehend better topics, such as videos and other digital web resources in place of traditional learning content. The online platform has gained importance recently due to the COVID-19 pandemic. This panic and the accessibility to internet facilities lead to various online education programs on platforms such as Zoom or Google Classroom. Progressive learning on such platforms allows students to move away from their workstations and learn independently. Students can study actively and participate in experimental learning with this freedom [10].

Implementing interactive teaching methods in periodontics enhances students’ critical thinking skills and improves their curiosity and logical reasoning in simplifying complex subject matters [11]. Furthermore, interactive education increases interaction and permits users to participate in the information, making it a more active, student-centered model. Another retrospective study among postgraduate dental students suggested that interactive teaching methods considerably improved the students’ academic performance [12]. Similarly, active learning strategies have been demonstrated to promote learning and understanding in subjects such as preclinical endodontics, forensic odontology, and pathology [13,14,15].

In addition, learning through e-classes was equally effective compared to conventional classroom teachings among medical students [16,17]. A cross-sectional study assessed the utility of online teaching among dental students, and it was found that the majority of the students liked online teaching [18]. However, the comparison of academic performance of interactive teaching methods in offline and online platforms in undergraduate dental programs has not been studied. This research was undertaken to assess whether there is any difference in academic performance of interactive teaching methods in offline and online platforms in Periodontics among undergraduate dental students at a dental college in India.

## 2. Materials and Methods

This study was approved by the Institutional Review Board of Saveetha Dental College and Hospitals, Tamil Nadu, India. This prospective study was conducted at Saveetha Dental College and Hospitals, Tamil Nadu, India, where the subject Periodontics was taught in interactive teaching method in an offline and online platform for undergraduates of the final year based on Bachelor of Dental Surgery curriculum.

The current study enrolled a total of 56 students. The students were given the option to choose either an online or offline class. Among 56 students, 27 students opted for an online class through Zoom (Group I), and 29 students opted for an offline class (Group II). The teaching plan was created for 1 week. The subject was divided into three modules: Module 1—Introduction and Etiopathogenesis of periodontal disease; Module 2—Diseases of the periodontium; and Module 3—Diagnosis and Treatment. Each module was further subdivided into 12 lectures.

Module 1 was discussed under introduction, gingiva, periodontal ligament, cementum, alveolar bone, age changes of the periodontium, dental plaque and calculus, influence of systemic diseases on the periodontium, environmental and genetic factors, iatrogenic factors, microbiology, and immunology. Module 2 was divided into the classification of gingival diseases, classification of periodontal diseases, stages of gingival inflammation, clinical features of gingivitis, gingival enlargement, acute gingival lesions, abscesses of the periodontium, periodontal pocket, periodontitis, necrotizing ulcerative conditions, patterns of bone loss, and the role of occlusion in periodontal disease. Module 3 was taught under the following headings: risk factors, prognosis, conventional diagnostic methods, advanced diagnostic methods, instruments and instrumentation, non-surgical periodontal therapy, gingival surgical procedures, periodontal flap surgery, regenerative periodontal therapy, resective periodontal therapy, furcation involvement and management, and mucogingival surgeries.

An interactive method of teaching was used for each module. Offline teaching consisted of lectures on specific topics, which lasted 20 min. Each lecture was followed by in-class activities such as concept mapping, quizzes, role playing, puzzle, and crosswords which lasted for 20 to 40 min. The online class was conducted via Zoom classroom, and it also consisted of lectures followed by activities. In order to avoid bias, similar activities were given to the students who opted for an online platform. It was made sure that the same interactive teaching method was conducted online for Group I students. All 3 modules were completed in 36 h for both groups. The topics covered, teaching methods, lectures, and activities were identical for both sets of students.

During each module, written tests were conducted for both the groups and were scored separately and added together to obtain a cumulative formative assessment. The students in groups I and II were given a summative score based on their performance on the written exam at the end of the module. (Figure 1). All the exam time limits were constant for both groups. In addition, the exams were monitored throughout the session.

The same examiner carried out both summative and formative assessments for groups I and II. In addition, the university’s third-year marks were obtained to avoid bias about the academic prospects of the two groups. The Kolmogorov–Smirnov test and the Shapiro–Wilk test of normality were used to evaluate the results. The results followed a parametric distribution according to the data. An unpaired *t*-test was used to compare the two groups’ scores on the summative and formative exams. Statistical Package for Social Sciences (SPSS Software, Version 23.0; IBM Corp., Armonk, NY, USA) was used to analyze the data. When the *p*-value was <0.05, the results were considered statistically significant.

## 3. Results

In the present study, 27 students opted for an online class through Zoom (Group I), and 29 students opted for offline class (Group II). Three students from Group I and four students from Group II failed to attend any of the modules or exams conducted during the module. Those students were excluded from the final data analysis. For statistical analysis, 24 students from Group I and 25 students from Group II were considered (Figure 2).

Students in both groups were compared based on formative and summative assessments. The unpaired *t*-test revealed no significant difference between the students’ academic performance in both groups during their third year (*p* = 0.423) (Table 1).

The independent *t*-test was used to compare both groups’ formative assessment scores during the module. We discovered no significant difference (*p* = 0.98) between groups. The summative evaluation scores of both groups were compared using an independent *t*-test. No difference (*p* = 0.93) between groups were observed (Table 2).

## 4. Discussion

The present study assessed the academic performance of interactive teaching methods in offline and online platforms in Periodontics among undergraduate dental students. For both groups of students, didactic and non-didactic teaching methods were implemented. The same teaching method was conducted online for the Group I students and in the classroom for the Group II students. According to the results, the performance was similar between the two groups in terms of formative and summative evaluation marks.

Several studies have demonstrated that interactive teaching mode was as excellent as conventional teaching methods, and in certain trials, it was proven to be the most effective learning approach [13,14,15,16,17,18,19,20,21,22]. Compared to traditional teaching approaches, interactive teaching has a considerable impact on cognitive achievement and learning attitude. In addition, for a wide range of learning outcomes, interactive teaching approaches have repeatedly been demonstrated to be equally as effective and, in some cases, more effective than traditional methods. A study by Veeraiyan DN et al. introduced the Multiple Interactive Learning Algorithm (MILA) in teaching Periodontics among undergraduate dental students. The study revealed that implementing interactive teaching methods that include a lecture followed by a game-based learning activity enhances the students’ performance [11]. In another study, video-based learning was implemented for one group of students, and another group of students received video-based lectures along with in-class activities. It was suggested that the blended module-based teaching resulted in significant improvement in the in-course assessments [13]. Our results are similar as the students in both groups showed improved performance in examinations, suggesting that the interactive teaching method was effective irrespective of the mode of teaching.

When students’ progress was compared with and without using technology throughout teaching, no difference was found between the classroom and distance learning groups [16]. Singh K et al. assessed the merits and demerits of online classes among dental students [17]. Most participants reported that they were allowed to interact and clarify their doubts with the teacher than they experienced in the actual classroom. In addition, an equal number of students thought both the actual classroom and the e-classroom were effective. The responses were also consistent across semesters. In addition, there was no significant difference in average marks between the two groups when structured interactive lectures were compared to traditional lectures as a teaching approach for pharmacology. However, a questionnaire analysis of the students’ perceptions revealed that they preferred the structured interactive teaching technique. Furthermore, interactive approaches and strategies such as flipped classrooms and multiple-choice questions in interactive mode engage students in the learning process, allowing them to retain more information and hence feel pleased [18].

Bains M et al., in their study, assessed the acceptance of didactic and non-didactic interactive methods of learning in the classroom and in online platforms among dental undergraduates in learning cephalometric tracing in Orthodontics. The findings revealed that students favored interactive teaching methods, implying that both online and offline sessions were practical and well-received [23]. This is in accordance with the present study, as the interactive mode of teaching improved the academic performance of the students in both the groups. In addition, Ochoa JG et al. suggested that an interactive style incorporating Web technology improves seizure disorder learning, maybe by stimulating critical thinking and increasing student motivation [24]. Similar results were obtained when the effectiveness of interactive teaching using media was assessed for teaching interpretation of arterial blood gas to medical students was compared to traditional lecture-based models [25]. Our findings are in agreement with the previous studies. In addition, our study finding highlights that interactive teaching method in both offline and online platforms resulted in significant growth in competence on the topics covered in all the three modules.

In addition, studies have demonstrated that e-learning may promote learning comparable to classroom lectures [26,27]. Another cross-sectional study evaluated the merits of e-learning among medical postgraduates after one month of online teaching, and it was suggested that e-learning is a viable alternative to classroom learning [28]. Similarly, studies by Bischoff W R et al. [29] and Gragan M K et al. [30] reported that there was no difference between traditional classroom learning and e-learning in terms of performance. Overall, our findings follow that of other studies.

The student’s intellectual ability might be a confounding factor; however, there was no significant difference between the two students’ third-year marks. In addition, the question paper used during the module and at the end of the module may not be a confounding factor because the questions were carefully designed for both sets of students to be of the same difficulty level.

The study design may be a potential limitation of the research. The study is not a randomized trial because students were given the option of learning in one of two platforms. Other flaws include the small sample size and a single study center, restricting the generalizability of the findings. More randomized clinical studies in a broader context are needed to obtain more data on the efficacy of various teaching methods.

## 5. Conclusions

This study found that interactive teaching methods in both offline and online platforms in Periodontics resulted in equivalent performance by the undergraduate dental students. Both the groups were benefited equally by interacting teaching method. Therefore, interactive teaching methods, either offline or online, provide students a beneficial learning environment.

## Figures and Tables

**Figure 1 ijerph-19-03170-f001:**
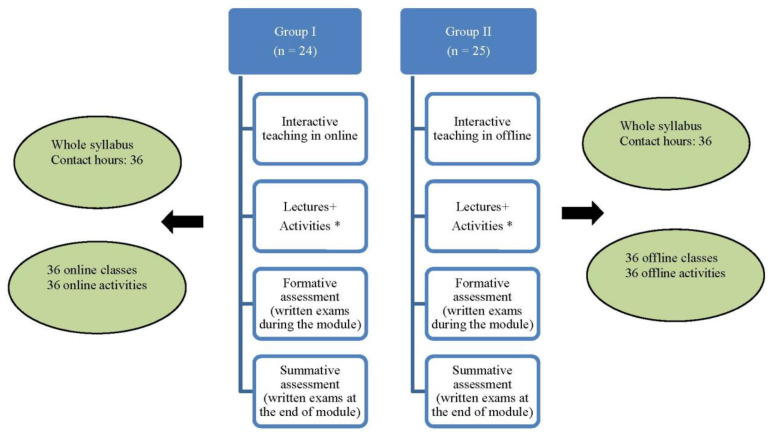
Schematic representation of interactive teaching in the online and offline platform. * 20 min lecture and 20–40 min activity from 8 a.m.–3 p.m. for 1 week.

**Figure 2 ijerph-19-03170-f002:**
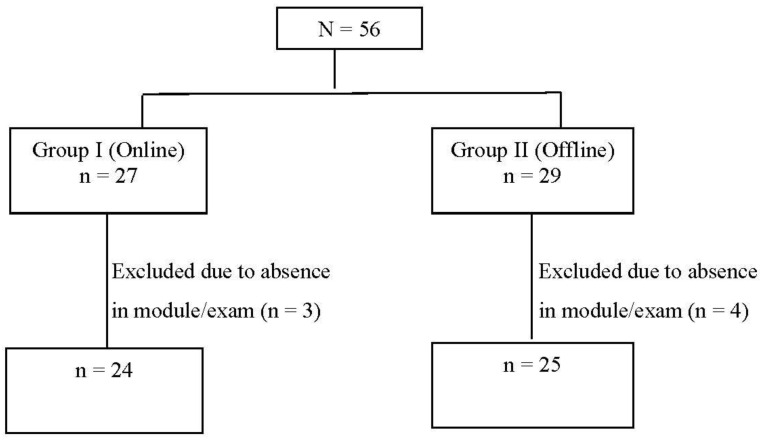
Study flow chart.

**Table 1 ijerph-19-03170-t001:** Comparison of third-year examination marks between the two groups.

Variables	Groups	Mean ± SD	*t*-Value	*p*-Value
Examination marks	Group I	74.25 ± 10.71	−0.808	0.423
	Group II	76.64 ± 10.003		

**Table 2 ijerph-19-03170-t002:** Comparison of formative and summative assessment marks between the two groups.

Variables	Groups	Mean ± SD	*t*-Value	*p*-Value
Formative assessment marks	Group I	77.88 ± 12.89	0.018	0.98
	Group II	77.80 ± 16.09		
Summative assessment marks	Group I	80.54 ± 8.39	0.090	0.93
	Group II	80.28 ± 11.57		

## Data Availability

Any data related to the study can be provided by the authors on reasonable request.

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
