# Peer review of "Comparison of Interactive Teaching in Online and Offline Platforms among Dental Undergraduates"

_ijerph, 2022, doi:10.3390/ijerph19063170_

Round 1

Reviewer 1 Report

This paper is a readable and concise description of work related to the pandemic. There are just a few suggestions for the authors.

The main issue is the Introduction section. The opening paragraph is very general. What is its purpose? The second paragraph is a little unstructured. What evidence is there for shorter attention spans? How does student-centered learning derive from the "digital evolution"? One recommendation is to just start with the third paragraph. Note also the paragraph on e-learning is out of place, should be moved earlier.

Relatedly, the paragraph beginning "The literature search revealed" needs to be better introduced; it comes at the reader from nowhere. Up until this point the overview has been quite generic and there has been no mention of periodontics. The literature search itself is fine and helps introduce the topic of this research. The last paragraph of the intro is good. However, the content under Discussion should be brought forward into the Intro.

The remainder of the paper is fine. The images are helpful, though in the methods there is slight confusion, "The online class...consisted of lectures followed by in-class activities". How were they in-class?

The conclusions are not very strong, merely that there were no major differences between students in the online and offline groups. The authors might consider implications of these findings.

Author Response

Reviewer 1:

This paper is a readable and concise description of work related to the pandemic. There are just a few suggestions for the authors.

Response: We would like to thank the academic editor and reviewers for taking their precious time to review this manuscript and give us comments. We would like to explicitly state that we agree with all the comments as these helped us improve the quality of our paper. We have made a conscious effort to answer all the remarks in the paper as advised by the reviewers and highlighted changes with red colour in the revised manuscript for their convenience. Changes have been carried out in the revised manuscript

  1. The main issue is the Introduction section. The opening paragraph is very general. What is its purpose? The second paragraph is a little unstructured. What evidence is there for shorter attention spans? How does student-centered learning derive from the "digital evolution"? One recommendation is to just start with the third paragraph. Note also the paragraph on e-learning is out of place, should be moved earlier.

Response: Thank you for your insightful suggestions and comments; corrections have been carried out in the introduction section in the revised manuscript as per your suggestions.

  1. Relatedly, the paragraph beginning "The literature search revealed" needs to be better introduced; it comes at the reader from nowhere. Up until this point the overview has been quite generic and there has been no mention of periodontics. The literature search itself is fine and helps introduce the topic of this research. The last paragraph of the intro is good. However, the content under Discussion should be brought forward into the Intro.

Response: Thank you for your insightful suggestions and comments; corrections have been carried out in the introduction section in the revised manuscript as per your suggestions.

  1. The remainder of the paper is fine. The images are helpful, though in the methods there is slight confusion, "The online class...consisted of lectures followed by in-class activities". How were they in-class?

Response: Thank you for your insightful suggestions and comments; the online class was conducted in Zoom classroom, and it also consisted of lectures followed by activities. In order to avoid bias, similar activities were given to the students who opted for an online platform. Corrections has been carried out in the revised manuscript.

  1. The conclusions are not very strong, merely that there were no major differences between students in the online and offline groups. The authors might consider implications of these findings.

Response: Thank you for your insightful suggestions and comments; corrections have been carried out in the revised manuscript as per your suggestions.

Reviewer 2 Report

The research question is not explicitly described in the introduction or in the research design section. Apparently, the only data analyzed are the results of student assessments, it appears to be a little poor base to demonstrate that both online and offline teaching are equivalent, that is the supposed research question I deduce from the paper.

The other elements in the discussion are based on literature citation and the link with the data presented are not clear.

I suggest reviewing the paper starting with the research question/s, explaining the research design and write a proper discussion and conclusion based on research data and design.

Author Response

Reviewer 2:

  1. The research question is not explicitly described in the introduction or in the research design section. Apparently, the only data analyzed are the results of student assessments, it appears to be a little poor base to demonstrate that both online and offline teaching are equivalent, that is the supposed research question I deduce from the paper.

Response: We would like to thank the academic editor and reviewers for taking their precious time to review this manuscript and give us comments. We would like to explicitly state that we agree with all the comments as these helped us improve the quality of our paper. We have made a conscious effort to answer all the remarks in the paper as advised by the reviewers and highlighted changes with red colour in the revised manuscript for their convenience. Changes have been carried out in the revised manuscript in the introduction section as per your suggestions.

  1. The other elements in the discussion are based on literature citation and the link with the data presented are not clear.

Response: Thank you for your insightful suggestions and comments; corrections has been carried out other study results were analysed with the results of the present study in the revised manuscript as per your suggestions.

  1. I suggest reviewing the paper starting with the research question/s, explaining the research design and write a proper discussion and conclusion based on research data and design.

Response: Thank you for your insightful suggestions and comments; corrections has been carried out in the discussion and conclusion sections in the revised manuscript as per your suggestions.

Reviewer 3 Report

I was pleased to review the paper under the title "COMPARISON OF INTERACTIVE TEACHING IN ONLINE AND OFFLINE PLATFORM AMONG UNDERGRADUATE DENTAL STUDENTS".
Although the topic of e-learning is especially important today, this study does not bring any novelty in this area. The study does not have a satisfactory methodology and cannot be published in this form.

INTRODUCTION:
Too extensive and largely unnecessary. A few paragraphs are more relevant to the discussion section. Plenty of sentences without confirmation (no proper references for claims).

MATERIAL AND METHODS:
It seems that the students are randomized into two almost equal groups. The methodology itself is not satisfactory. There are no innovations that have been applied in the teaching process itself…

RESULTS
Few variables have been analyzed, so the results are scarce.

DISCUSSION:
It all comes down to a lack of results, and thus the whole discussion is over describing other studies and the application of e-learning. There is no clear discussion of the material and results of this study.

CONCLUSION:
Short, and considering the methodology, sufficient.

Author Response

Reviewer 3:

  1. I was pleased to review the paper under the title "COMPARISON OF INTERACTIVE TEACHING IN ONLINE AND OFFLINE PLATFORM AMONG UNDERGRADUATE DENTAL STUDENTS".
    Although the topic of e-learning is especially important today, this study does not bring any novelty in this area. The study does not have a satisfactory methodology and cannot be published in this form.

Response: We would like to thank the academic editor and reviewers for taking their precious time to review this manuscript and give us comments. We would like to explicitly state that we agree with all the comments as these helped us improve the quality of our paper. We have made a conscious effort to answer all the remarks in the paper as advised by the reviewers and highlighted changes with red colour in the revised manuscript for their convenience. Changes have been carried out in all the sections in the revised manuscript.

In the introduction, necessary corrections were made, and references were added. Even though there are studies based on interactive teaching methods, no studies compare interactive teaching methods in online and offline mode. Therefore, this study methodology was designed in such a way to compare the interactive teaching method in online and offline mode. This study focussed only on academic performance; therefore, summative and formative assessments were considered. Also, in the discussion, other study results were analyzed with the present study results.

  1. INTRODUCTION:
    Too extensive and largely unnecessary. A few paragraphs are more relevant to the discussion section. Plenty of sentences without confirmation (no proper references for claims).

Response: Thank you for your insightful suggestions and comments; necessary corrections were made in the introduction, and references were added in the revised manuscript as per your suggestions.

  1. MATERIAL AND METHODS:
    It seems that the students are randomized into two almost equal groups. The methodology itself is not satisfactory. There are no innovations that have been applied in the teaching process itself…

Response: Thank you for your insightful suggestions and comments; no studies compare interactive teaching methods online and offline. Therefore, this study methodology was designed in such a way to compare the interactive teaching method in online and offline mode. Changes have been carried out in the revised manuscript as per your suggestions

  1. RESULTS
    Few variables have been analyzed, so the results are scarce.

Response: Thank you for your insightful suggestions and comments; this study focussed only on academic performance; therefore, summative and formative assessments were considered. Also, in the discussion, other study results were analyzed with the present study results. Changes have been carried out in the revised manuscript as per your suggestions

  1. DISCUSSION:
    It all comes down to a lack of results, and thus the whole discussion is over describing other studies and the application of e-learning. There is no clear discussion of the material and results of this study.

Response: Thank you for your insightful suggestions and comments; corrections have been carried out in the discussion section of the revised manuscript as per your suggestions.

  1. CONCLUSION:
    Short, and considering the methodology, sufficient.

Response: Thank you for your insightful suggestions and comments; corrections have been carried out in the conclusion section of the revised manuscript as per your suggestions.

Reviewer 3:

  1. I was pleased to review the paper under the title "COMPARISON OF INTERACTIVE TEACHING IN ONLINE AND OFFLINE PLATFORM AMONG UNDERGRADUATE DENTAL STUDENTS".
    Although the topic of e-learning is especially important today, this study does not bring any novelty in this area. The study does not have a satisfactory methodology and cannot be published in this form.

Response: We would like to thank the academic editor and reviewers for taking their precious time to review this manuscript and give us comments. We would like to explicitly state that we agree with all the comments as these helped us improve the quality of our paper. We have made a conscious effort to answer all the remarks in the paper as advised by the reviewers and highlighted changes with red colour in the revised manuscript for their convenience. Changes have been carried out in all the sections in the revised manuscript.

In the introduction, necessary corrections were made, and references were added. Even though there are studies based on interactive teaching methods, no studies compare interactive teaching methods in online and offline mode. Therefore, this study methodology was designed in such a way to compare the interactive teaching method in online and offline mode. This study focussed only on academic performance; therefore, summative and formative assessments were considered. Also, in the discussion, other study results were analyzed with the present study results.

  1. INTRODUCTION:
    Too extensive and largely unnecessary. A few paragraphs are more relevant to the discussion section. Plenty of sentences without confirmation (no proper references for claims).

Response: Thank you for your insightful suggestions and comments; necessary corrections were made in the introduction, and references were added in the revised manuscript as per your suggestions.

  1. MATERIAL AND METHODS:
    It seems that the students are randomized into two almost equal groups. The methodology itself is not satisfactory. There are no innovations that have been applied in the teaching process itself…

Response: Thank you for your insightful suggestions and comments; no studies compare interactive teaching methods online and offline. Therefore, this study methodology was designed in such a way to compare the interactive teaching method in online and offline mode. Changes have been carried out in the revised manuscript as per your suggestions

  1. RESULTS
    Few variables have been analyzed, so the results are scarce.

Response: Thank you for your insightful suggestions and comments; this study focussed only on academic performance; therefore, summative and formative assessments were considered. Also, in the discussion, other study results were analyzed with the present study results. Changes have been carried out in the revised manuscript as per your suggestions

  1. DISCUSSION:
    It all comes down to a lack of results, and thus the whole discussion is over describing other studies and the application of e-learning. There is no clear discussion of the material and results of this study.

Response: Thank you for your insightful suggestions and comments; corrections have been carried out in the discussion section of the revised manuscript as per your suggestions.

  1. CONCLUSION:
    Short, and considering the methodology, sufficient.

Response: Thank you for your insightful suggestions and comments; corrections have been carried out in the conclusion section of the revised manuscript as per your suggestions.

Round 2

Reviewer 3 Report

I thank the authors for the revised article. However, concerns remain about the methodology itself. I stand that this paper cannot be accepted for publication.

Author Response

Comments and Suggestions for Authors

I thank the authors for the revised article. However, concerns remain about the methodology itself. I stand that this paper cannot be accepted for publication.

Dear Prof,

We want to thank the editor and the reviewers for taking their precious time to review this manuscript and give us their comments. We would like to explicitly state that we agree with all the comments as these helped us improve the quality of our paper. We have made a conscious effort to answer all the remarks in the paper as you and the reviewer insightful comments and highlighted changes with red colour in the revised manuscript for their convenience. Changes have been carried out in the revised manuscript

The aim of the present study was to assess the academic performance of interactive teaching methods in offline and online platforms. Therefore, the methodology was designed in a way to assess the academic performance and hence formative and summative assessment marks were considered as parameters.

Also, discussion and conclusion were revised.